# Prevalence and Factors Associated with Prosthetic Joint Infections in Patients with *Staphylococcus aureus* Bacteraemia: A 7-Year Retrospective Study

**DOI:** 10.3390/antibiotics11101323

**Published:** 2022-09-28

**Authors:** Matthaios Papadimitriou-Olivgeris, Laurence Senn, Claire Bertelli, Bruno Grandbastien, Sylvain Steinmetz, Noémie Boillat-Blanco

**Affiliations:** 1Infectious Diseases Service, Lausanne University Hospital and University of Lausanne, 1011 Lausanne, Switzerland; 2Infection Prevention and Control Unit, Lausanne University Hospital and University of Lausanne, 1011 Lausanne, Switzerland; 3Institute of Microbiology, Lausanne University Hospital and University of Lausanne, 1011 Lausanne, Switzerland; 4Department of Orthopaedics and Traumatology, Lausanne University Hospital and University of Lausanne, 1011 Lausanne, Switzerland

**Keywords:** *Staphylococcus aureus*, bloodstream infection, prosthetic joint infection, arthroplasty, community-acquired infection

## Abstract

Background: *Staphylococcus aureus* is the main cause of haematogenous prosthetic joint infections (PJI). We aimed to describe the prevalence and factors associated with PJI in patients with documented *S. aureus* bacteraemia. Methods: Adult patients with *S. aureus* bacteraemia and presence of prosthetic joint hospitalized in Lausanne University Hospital during a seven-year period (2015–2021) were included. Results: Among 135 patients with *S. aureus* bacteraemia and prosthetic joints, 38 (28%) had PJI. Multivariate analysis revealed that the presence of PJI was associated with knee arthroplasty (*P* 0.029; aOR 3.00, 95% CI 1.12–8.05), prior arthroplasty revision (*P* 0.034; aOR 3.59, 95% CI 1.10–11.74), community-acquired bacteraemia (*P* 0.005; aOR 4.74, 95% CI 1.61–14.01) and age < 70 years (*P* 0.007; aOR 9.39, 95% CI 1.84–47.85). Conclusions: PJI was common among patients with documented *S. aureus* bacteraemia. PJI was associated with characteristics of the prosthesis, such as prior arthroplasty revisions and knee prosthesis.

## 1. Introduction

Prosthetic joint infection (PJI) is a significant complication of arthroplasties, leading to multiple admissions, increased healthcare costs and associated with increased morbidity and mortality [1,2,3]. Although the risk of PJI after arthroplasty remains low (up to 2.2%), the increase in PJIs can be attributed to the continual rise in number of total joint arthroplasties and an older and more comorbid population [4,5]. Several risk factors for PJI development after total joint arthroplasty are identified and include increased body mass index, diabetes mellitus, immunosuppression, knee arthroplasty and prior joint surgery [6,7].

A significant proportion of PJI are due to haematogenous seeding usually occurring more than 3 months after arthroplasty [8,9]. Haematogenous PJI presents as an abrupt onset of joint symptoms following a symptom-free period, ranging from 3 months to several years. Such symptoms include joint pain, erythema, swelling and drainage [1]. An increase in bacteraemic PJIs has been observed over the last three decades [5]. *Staphylococcus aureus* is the most common pathogen implicated in haematogenous PJIs [9]. In patients with prosthetic joints and occurrence of *S. aureus* bacteraemia, the prevalence of PJI ranges between 20% and 42%, but these results are mostly from studies with low patient numbers [8,10,11,12].

The aim of this study was to describe the prevalence and factors associated with PJI in patients with documented *S. aureus* bacteraemia in a Swiss university hospital.

## 2. Results

A total of 151 patients with *S. aureus* bacteraemia and presence of a prosthetic joint were identified, from which 16 were excluded (bacteraemia occurrence within 3 months after prosthesis implantation or revision). Hence, 135 patients with 208 joint prostheses (117 hip, 89 knee and 2 shoulder; 73 patients had more than one prosthesis) were included. Of these 208 prostheses, 24 had prior revision; 22 for mechanical reasons (loosening of the prosthesis without signs of infection, metallosis, etc.) and two for PJI by another pathogen.

A total of 39 (19%) PJI cases were diagnosed at the time of *S. aureus* bacteraemia in 37 (25%) patients (two patients had two PJIs at the same time). Only one patient (3%) was not diagnosed clinically with a PJI at the time of *S. aureus* bacteraemia; indeed, a collection on CT scan was documented 5 days after *S. aureus* bacteraemia. Perioperative cultures were positive for *S. aureus* in 37 (93%) cases and 36 (95%) patients. Orthopaedic characteristics of the joint arthroplasties among patients with PJIs and those without are shown in Table 1.

Among patients with PJI, the median duration of local symptoms before documentation of *S. aureus* bacteraemia was 3 days (range: 1–40 days), while the median duration of fever (among 33 patients with fever) was 2 days (range: 1–3). Joint pain was present in all 39 initially symptomatic prostheses (Table 1). Swelling was present in 20 (51%) prostheses, periarticular warmth in 13 (33%), periarticular erythema in 10 (26%), and drainage in one (3%). Among patients without PJI, 18% of prosthetic joints presented at least one symptom, with joint pain being the most prominent. Upon bacteraemia onset, no patient was actively receiving antibiotic treatment.

Among the 97 patients who did not have PJI diagnosed at the time of *S. aureus* bacteraemia, 6-month follow-up information was available for 64 patients (66%) of whom only one developed *S. aureus* PJI (at 98 days from *S. aureus* bacteraemia) and 20 patients died in the interval (range 3–131 days after *S. aureus* bacteraemia) without sign of PJI. For 13 patients, the last follow-up occurred earlier (range 24–124 days after *S. aureus* bacteraemia).

PJI upon initial hospitalization was associated with younger age, community-acquired bacteraemia, presence of knee prosthesis, prior revision, higher CRP and lower rate of malignancy (Table 2). On multivariate analysis, PJI was associated with knee arthroplasty (*P* 0.029; aOR 3.00, 95% CI 1.12–8.05), prior arthroplasty revision (*P* 0.034; aOR 3.59, 95% CI 1.10–11.74), community-acquired bacteraemia (*P* 0.005; aOR 4.74, 95% CI 1.61–14.01) and age < 70 years (*P* 0.007; aOR 9.39, 95% CI 1.84–47.85).

## 3. Discussion

Our results showed that PJI diagnosis is frequent among patients with *S. aureus* bacteraemia in the presence of prosthetic joints (28%); this rate is similar to that reported in the literature (20–42%) [8,10,11]. This diagnosis must always be sought in the absence of another obvious source.

Consistent with previous studies, community as opposed to nosocomial acquisition of *S. aureus* bacteraemia is associated with PJI [10,11,13]. The duration of nosocomial bacteraemia before diagnosis and treatment initiation is probably most often short and prevents such a complication. In this study, age < 70 years was also associated with PJI. This is in line with a previous study which showed that among patients with bacteraemia, patients with PJI were younger than those without [13]. However, most studies did not show any age difference [8,10,11]. Younger patients were closer to the date of last revision (median of 66 vs. 74 months; *P* 0.584); in previous studies PJI related to bacteraemia was higher during the first year after operation (implantation or revision) [8,14]. Another explanation could be the type of prosthesis implanted, but such information was not collected.

Prosthetic joint characteristics play an important role in PJI development. Indeed, prior revision (more than 3 months before infection) was associated with *S. aureus* bacteraemic PJI. Although prior revision was previously reported [11,13,15], this association with PJI was not universally found [8,10]. Another prosthesis characteristic associated with PJI was presence of a knee prosthesis; such an association was found in some, but not all previous studies with haematogenous *S. aureus* PJIs [4,13,16]. Knee prostheses are usually more difficult to implant than hip prostheses and knee arthroplasty usually results in bone loss, which necessitates a larger prosthesis to compensate for the loss and in turn increases the prosthetic material surface prone to bacterial adhesion [15]. In contrast to Tande et al. [11], the presence of multiple joint prostheses was not associated with PJI in the present study.

Only a minority of PJI were diagnosed in a second step, which reinforces previously published data that in the absence of clinical signs of PJI at the time of *S. aureus* bacteraemia, the risk of PJI is extremely low, and no additional diagnostic investigations are necessary [11,17]. As previously shown, the most prominent local symptom was pain, which was universally present in our study, followed by joint swelling, warmth and erythema [11]. As previously shown, in patients without diagnosis of PJI a small percentage of prostheses will become symptomatic upon bacteraemia, with the most prominent symptom being joint pain [11].

Only one patient (2%) who did not have PJI diagnosed at the time of *S. aureus* bacteraemia (follow-up for 6 months) developed metachronous PJI (124 days after *S. aureus* bacteraemia). This finding was in line with previous studies that showed low risk for metachronous PJI. Tande et al. found four (8%) metachronous PJI among 50 patients in 3.4 years follow-up (174 to 670 days after *S. aureus* bacteraemia). This higher percentage can be explained by the longer follow-up period (3.4 years); indeed, among these four PJIs only one occurred within 6 months from *S. aureus* bacteraemia [11]. In another study, no metachronous infection was diagnosed in 19 patients within a 12-month follow-up after *S. aureus* bacteraemia [10].

The present study has several limitations. First, it was a single centre retrospective study with a limited number of patients; in contrast to previous studies [4,10,11,13], upon *S. aureus* bacteraemia all patients were evaluated by a specialized team, thus limiting the number of undiagnosed cases. However, the study size is comparable to the largest study to date [13] and significantly higher than most previous studies [8,10,11]. Second, we may have missed PJIs that occurred at a later stage as 6-month follow-up information was missing for 13% of the patients. Third, the prevalence of PJIs among patients with *S. aureus* bacteraemia and presence of prosthetic joint could be lower in other settings, since our centre is a tertiary referral hospital. Last, the type of prosthesis implanted was not collected, thus its impact was not assessed.

## 4. Materials and Methods

This was a retrospective study conducted at Lausanne University Hospital, Lausanne, Switzerland over a seven-year period (1st January 2015 to 31st December 2021). Lausanne University Hospital is a 1100-bed primary and tertiary care hospital and has a dedicated Septic Surgery Unit.

Inclusion criteria were adults (age ≥ 18 years), *S. aureus* bacteraemia and the presence of at least one prosthetic joint. Exclusion criteria were patients’ written refusal of the use of data and bacteraemia occurrence within 3 months after prosthetic joint implantation or revision (considered to have primary postsurgical PJI) [1,9].

We decided on a 3-month cut-off for the characterization of an *S. aureus* PJI as primary versus haematogenous based on previous studies [1,9]. Since *S. aureus* is highly virulent, its perioperative inoculation leads to local signs development within days or weeks from joint implantation or revision, and rarely after 3 months.

*S. aureus*-positive blood cultures were extracted from the database of the microbiological laboratory. Blood cultures were collected during routine care. Information on the presence or absence of a prosthetic joint was extracted from the patients’ electronic health records.

Data regarding demographics (age, sex), comorbidities (arterial hypertension, coronary disease, congestive heart failure, chronic obstructive pulmonary disease, cirrhosis, diabetes mellitus, chronic kidney disease, malignancy, obesity, autoimmune disease, immunosuppression), number of positive blood cultures, polymicrobial bacteraemia, persistent bacteraemia, C-reactive protein, presence of sepsis or septic shock, prostheses characteristics (timing of implantation, site of prosthetic joint, prior revision), presence of joint symptoms or fever were retrieved from patients’ electronic health records. Study data were collected and managed using REDCap by an infectious disease specialist. REDCap electronic data capture tools is hosted at Lausanne University Hospital. REDCap (Research Electronic Data Capture) is a secure, web-based software platform designed to support data capture for research studies [18,19].

The date of collection of the first positive blood culture was defined as infection onset. Bacteraemia was characterized as community-acquired according to Friedman et al. [20]. PJI was defined upon initial hospitalization according to the European Bone and Joint Infection Society [16]. Diagnosis of PJI the time of bacteraemia was based on at least one positive intraoperative sample (synovial fluid or tissue culture), positive aspiration fluid culture and cytologic/histologic findings. Infection was categorized as sepsis or septic shock according to the definition proposed by the Sepsis-3 International Consensus [21]. C-reactive protein level with a cut-off of ≥220 mg/L was used, because it was previously found to be associated with PJI among patients with *S. aureus* bacteraemia [13]. Malignancy was considered as active solid tumour or haematologic cancer or during the first year from complete remission. Patients’ records were reviewed until the latest clinical visit or death to assess for the occurrence of *S. aureus* PJI not diagnosed at the time of initial hospitalization.

SPSS 26.0 (SPSS, Chicago, IL, USA) was used for data analysis. Categorical variables were depicted as counts and percentages, while continuous with medians and interquartile ranges (IQRs). Differences between patients with and without PJI were analysed by Fisher’s exact test and Mann–Whitney *U* test, as appropriate. We investigated factors associated with PJI with multivariate logistic regression using backward selection including non-collinear variables. All statistical tests were two-tailed and *P* < 0.05 was considered statistically significant.

## 5. Conclusions

PJI was common among patients with documented *S. aureus* bacteraemia, particularly for community-acquired bacteraemia, and almost always clinically conspicuous at the time of bacteraemia diagnosis. PJI was associated with characteristics of the prosthesis, such as prior arthroplasty revisions and knee prosthesis. The risk of metachronous PJI was low.

## Figures and Tables

**Table 1 antibiotics-11-01323-t001:** Orthopaedic characteristics of the joint arthroplasties.

	No PJI (*n* = 168)	PJI (*n* = 40)	*P*
**Localization of prosthesis**			
Hip	102 (61%)	15 (38%)	
Knee	64 (38%)	25 (63%)	0.007 ^1^
Other	2 (1%)	0 (0%)	
**History of revisions**			
None	156 (93%)	28 (70%)	
One or more	12 (7%)	12 (30%)	<0.001
**Months from last revision**	91 (52–168)	104 (19–161)	0.355
**Joint symptoms upon bacteraemia**			
Asymptomatic	137 (82%)	1 (3%)	<0.001
Joint pain	25 (15%)	39 (98%)	<0.001
Swelling	4 (2%)	20 (50%)	<0.001
Periarticular warmth	3 (2%)	10 (25%)	<0.001
Periarticular erythema	3 (2%)	13 (33%)	<0.001
Drainage	0 (0%)	1 (3%)	0.192

Data are depicted as number and percentage or median and Q1-3. ^1^ Knee arthroplasty compared to other sites (hip, shoulder).

**Table 2 antibiotics-11-01323-t002:** Characteristics of patients with *S. aureus* bacteraemia in the presence of prosthetic joint according to the presence or absence of prosthetic joint infection (PJI).

	Univariate Analysis			Multivariate Analysis	
	No PJI (*n* = 97)	PJI (*n* = 38)	*P*	aOR (95% CI)	*P*
Demographics					
Male sex	67 (69%)	26 (68%)	0.941		
Age (years)	78 (69–84)	71 (63–77)	0.001		
Age < 70 years old	27 (28%)	19 (50%)	0.025	9.39 (1.84–47.85)	0.007
Co-morbidities					
Arterial hypertension	58 (59%)	26 (68%)	0.352		
Coronary disease	27 (28%)	5 (13%)	0.071		
Congestive heart failure	8 (8%)	0 (0%)	0.105		
Chronic obstructive pulmonary disease	12 (12%)	2 (5%)	0.348		
Cirrhosis	11 (11%)	3 (8%)	0.757		
Diabetes mellitus	33 (34%)	9 (24%)	0.303		
Chronic kidney disease (moderate or severe)	27 (28%)	8 (21%)	0.419		
Malignancy (solid organ or haematologic)	21 (22%)	2 (5%)	0.023		
Obesity	27 (28%)	16 (42%)	0.109		
Autoimmune disease	15 (16%)	7 (18%)	0.676		
Immunosuppression	21 (22%)	4 (11%)	0.217		
Setting of infection onset					
Community-acquired	64 (66%)	37 (97%)	<0.001	4.74 (1.61–14.01)	0.005
Nosocomial	33 (34%)	1 (3%)			
Microbiological data					
Two or more blood cultures positive	75 (77%)	32 (84%)	0.482		
Polymicrobial bacteraemia	8 (8%)	1 (3%)	0.444		
Methicillin-resistance	11 (11%)	1 (3%)	0.178		
Persistent bacteraemia (≥48 h)	22 (23%)	14 (37%)	0.094		
Antibiotic treatment (upon bacteraemia onset)	0 (0%)	0 (0%)	-		
Clinical presentation					
Temperature (°C)	38.4 (38.0–39.0)	38.6 (37.8–39.0)	0.892		
Fever (temperature >38 °C)	81 (84%)	33 (87%)	0.630		
Sepsis	45 (46%)	12 (32%)	0.117		
Septic shock	14 (14%)	5 (13%)	1.000		
SOFA score	3 (1–5)	2 (1–4)	0.449		
Localization of prosthesis					
Hip	68 (70%)	19 (50%)			
Knee	42 (43%)	25 (68%)	0.020 ^1^	3.00 (1.12–8.05)	0.029 ^1^
Shoulder	2 (2%)	0 (0%)			
Multiple joint prosthesis	44 (45%)	29 (50%)	0.624		
Months since implantation	71 (37–139)	103 (53–143)	0.368		
Prior revision (more than 3 months before)	9 (9%)	12 (32%)	0.003	3.59 (1.10–11.74)	0.034
Months since last revision	71 (37–139)	84 (19–122)	0.845		
Laboratory data					
C-reactive protein (mg/L)	229 (118–307)	337 (243–388)	<0.001		
C-reactive protein ≥ 220 mg/L	50 (52%)	29 (76%)	0.011		

Data are depicted as number and percentage or median and Q1-3. ^1^ Knee arthroplasty compared to other sites (hip, shoulder); 95% CI: 95% confidence interval; aOR: adjusted odds ratio; SOFA: Sequential Organ Failure Assessment.

## Data Availability

The datasets generated during and/or analysed during the current study are available from the corresponding author on reasonable request.

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
