# Peer review of "Prevalence and Factors Associated with Prosthetic Joint Infections in Patients with Staphylococcus aureus Bacteraemia: A 7-Year Retrospective Study"

_antibiotics, 2022, doi:10.3390/antibiotics11101323_

Round 1

Reviewer 1 Report

Dear authors, 

my warmest thanks for an interesting analysis about PJI in SAB. I enjoyed reading your article; my only suggestions are typographical ones: Please use italics for latin genus- and species names consistently, and fix typos:

- "Sthaphylococcus" in line 35

- Superfluous point "." in line 69

Again, thank you for your work.

Kind regards

Author Response

Points raised by Reviewer 1

Point 1

Please use italics for latin genus- and species names consistently, and fix typos. - "Sthaphylococcus" in line 35.

Response

We applied italics in genus and species names throughout the text and fixed typos

Point 2

Superfluous point "." in line 69

Response

"." is erased

Reviewer 2 Report

This manuscript about PJI occurring in patient with S. aureus bacteraemia is of interest. It corroborates previous results and highlights other potential risks factors. Still, several points of improvement can be pointed out.

Overall comments:

-        Authors should specify the methods used to detect bacteraemia and PJI in patients. Were the diagnoses performed concomitantly?

-        The antibiotic treatments received by patients might had huge impact on the induction of bacteraemia and PJI. Do authors have information about prior antibiotics treatments in patients included?

-        The potential role of factors significantly impacting the PJI induction might be more discussed. For example, what may explain that patients with PJI were younger? Also, how malignancy can be linked to a lower risk of having PJI?

-        The reason for rejecting patients with bacteraemia occurrence within 3 months after prosthetic joint implantation or revision might be explained better. Also, the reference [3] itself refers to a publication of Zimmerli et al. (2004) to justify this 3-months delay. Thus, the original publicaiton should be cited.

-        Lines 73-75: “Indeed, prior revision (more than 3 months before infection) was associated with S. aureus bacteraemic PJI.” Do authors have information on the reason of these revisions? Were they linked to first PJI? If yes, this might have an important impact on the risk of bacteraemia/relapse.

Specific comments:

-        Italic is lacking on several bacterial names. The term “bacteraemia” is written with different orthographs. Some terms are incorrectly written (line 78: “protheses”).

-        The term “malignancy” should be specified, as it can applied to various pathologies.

-        Authors should specify the days (or at least the months) of the beginning and of the end of the inclusion period.

-        Lines 35-37: “In patients with S. aureus bacteraemia (SAB) and the presence of prosthetic joints, the risk of PJI is 20% to 42%”. The sentence is not clear and might be rephrased.

-        The design of table 1 might be improved: section titles and P < 0.05 can appear in bold.

-        Line 71: “However, most studies did show any age difference. [4-6]” It seems the sentence lack a “not”.

-        Line 85: A verb is lacking in the last part of the sentence.

Author Response

Points raised by Reviewer 2

Point 1

Authors should specify the methods used to detect bacteraemia and PJI in patients. Were the diagnoses performed concomitantly?

Response

We clarified the way bacteraemia were detected in the methods section.We elaborated on the timing of diagnosis of bacteraemia and PJI. As specified in the Results section, all but one patient were diagnosed concomitantly.

Point 2

The antibiotic treatments received by patients might had huge impact on the induction of bacteraemia and PJI. Do authors have information about prior antibiotics treatments in patients included?

Response

We have information on antibiotics use at the time of admission for patients with community-acquired bacteraemia and on antibiotic use during hospital stay for patients with bacteraemia during hospital stay. We added this information in Table 2.

Point 3

The potential role of factors significantly impacting the PJI induction might be more discussed. For example, what may explain that patients with PJI were younger? Also, how malignancy can be linked to a lower risk of having PJI?

Response

We extended the discussion on potential role of identified factors. Younger patients were closer to day of implantation/revision. Another explanation could be the type of prosthesis implanted, but such information was not collected. Concerning the malignancy, and after adjusting for age, malignancy was not associated anymore with PJI.

Point 4

The reason for rejecting patients with bacteraemia occurrence within 3 months after prosthetic joint implantation or revision might be explained better. Also, the reference [3] itself refers to a publication of Zimmerli et al. (2004) to justify this 3-months delay. Thus, the original publicaiton should be cited.

Response

We elaborated the use of the 3 months cutoff. Concerning the study cited, it was the study from Wouthuyzen-Bakker et al. [9]

Point 5

Lines 73-75: “Indeed, prior revision (more than 3 months before infection) was associated with S. aureus bacteraemic PJI.” Do authors have information on the reason of these revisions? Were they linked to first PJI? If yes, this might have an important impact on the risk of bacteraemia/relapse.

Response

Those revisions (24 joint prosthesis) were mainly due to mechanical reasons (loosening of the prothesis without signs of infection, metallosis, etc). Only two were revised due to PJI by another pathogen. This information was added in the results section.  

Point 6

Italic is lacking on several bacterial names. The term “bacteraemia” is written with different orthographs. Some terms are incorrectly written (line 78: “protheses”).

Response

Italics was applied in bacterial species ang genus. “Bacteraemia” was corrected throughout the text. “Protheses” is changed to “prostheses”.

Point 7

The term “malignancy” should be specified, as it can applied to various pathologies.

Response

We defined malignancy in the Materials and Methods.

Point 8

Authors should specify the days (or at least the months) of the beginning and of the end of the inclusion period.

Response

The inclusion period is specified (1st January 2015 to 31st December 2021) in the Materials and Methods. 

Point 9

Lines 35-37: “In patients with S. aureus bacteraemia (SAB) and the presence of prosthetic joints, the risk of PJI is 20% to 42%”. The sentence is not clear and might be rephrased

Response

We rephrased the sentence.  

Point 10

The design of table 1 might be improved: section titles and P < 0.05 can appear in bold.

Response

Different sections appear in bold. P< 0.05 are highlighted.

Point 11

Line 71: “However, most studies did show any age difference. [4-6]” It seems the sentence lack a “not”.

Response

You are right. The “not” is added.

Point 12

Line 85: A verb is lacking in the last part of the sentence.

Response

You are right. A verb is added.

Round 2

Reviewer 2 Report

The authors have done efforts to improve their manuscript, in accordance with our suggestions. Thank you.

Here are few remaining remarks: 

Authors’ response to point 4: “We elaborated the use of the 3 months cutoff. Concerning the study cited, it was the study from Wouthuyzen-Bakker et al. [9]”. The publication cited is indeed the one of from Wouthuyzen-Bakker et al. [9]. Our point was that Wouthuyzen-Bakker et al. themselves refers to Zimmerli et al. (2004) to justify the 3 months cutoff, thus the two publications may be cited.

Table 1: As in table 2, the particularity of statistical analysis (e.g. comparison hip/knee/other) should be explain at the bottom of the table.

Line 57: “bacteraemia” should not be in italic.

Line 167: “positive” should not be in italic.

Author Response

Point 1

Authors’ response to point 4: “We elaborated the use of the 3 months cutoff. Concerning the study cited, it was the study from Wouthuyzen-Bakker et al. [9]”. The publication cited is indeed the one of from Wouthuyzen-Bakker et al. [9]. Our point was that Wouthuyzen-Bakker et al. themselves refers to Zimmerli et al. (2004) to justify the 3 months cutoff, thus the two publications may be cited.

Response

You are right. Both studies are cited.

Point 2

Table 1: As in table 2, the particularity of statistical analysis (e.g. comparison hip/knee/other) should be explain at the bottom of the table.

Response

It is added

Point 3

Line 57: “bacteraemia” should not be in italic.

Response

You are right

Point 4

Line 167: “positive” should not be in italic.

Response

You are right